# Intravascular Lithotripsy as a Novel Treatment Method for Calcified Unprotected Left Main Diseases—Comparison to Rotational Atherectomy—Short-Term Outcomes

**DOI:** 10.3390/ijerph19159011

**Published:** 2022-07-25

**Authors:** Piotr Rola, Jan Jakub Kulczycki, Adrian Włodarczak, Mateusz Barycki, Szymon Włodarczak, Marek Szudrowicz, Łukasz Furtan, Artur Jastrzębski, Maciej Pęcherzewski, Maciej Lesiak, Adrian Doroszko

**Affiliations:** 1Faculty of Health Sciences and Physical Culture, Witelon Collegium State University, 59-220 Legnica, Poland; wlodarczak.adrian@gmail.com; 2Department of Cardiology, Provincial Specialized Hospital, 59-220 Legnica, Poland; mateusz.barycki@gmail.com (M.B.); lukas.furtan@gmail.com (Ł.F.); 3Department of Cardiology, The Copper Health Centre (MCZ), 59-300 Lubin, Poland; jan.jakub.kulczycki@gmail.com (J.J.K.); wlodarczak.szy@gmail.com (S.W.); marek.szudrowicz@gmail.com (M.S.); arias@poczta.onet.pl (A.J.); maciej999@gmail.com (M.P.); 41st Department of Cardiology, University of Medical Sciences, 61-848 Poznan, Poland; maciej.lesiak@skpp.edu.pl; 5Clinical Department of Internal Medicine and Occupational Diseases, Hypertension and Clinical Oncology, Faculty of Medicine, Wroclaw Medical University, 50-556 Wroclaw, Poland; adrian.doroszko@gmail.com

**Keywords:** left main, rotational atherectomy, intravascular lithotripsy, high-risk percutaneous coronary intervention, percutaneous coronary intervention (PCI), vascular disease, coronary artery diseases (CAD), cardiovascular diseases, shock wave intravascular lithotripsy device

## Abstract

Background: The unprotected calcified Left Main disease represents a high-risk subset for percutaneous coronary intervention (PCI), and it is associated with a higher number of periprocedural complications and an increased rate of in-stent thrombosis and restenosis. Adequate lesion preparation plays a crucial role in achieving a favorable PCI outcome. Rotational Atherectomy (RA) is a well-established plaque-modifying method; nevertheless, the data regarding the effectiveness of RA in LM diseases is scarce. Recently, the novel ShockWave-Intravascular-Lithotripsy(S-IVL) device has been introduced to the PCI armamentarium in order to modify the calcified plaque. Methods: We performed a retrospective evaluation of 44 consecutive subjects who underwent the LM-PCI, and who were supported by either the RA or S-IVL. Results: The Rota group consisted of 29 patients with a mean syntax score of 28.0 ± 7.5. The S-IVL group was composed of 15 subjects with a syntax score of 23.3 ± 13.0 There were no statistical differences regarding MACE between the RA and Shockwave arms of the in-hospital group (10.3% vs. 6.7%), or in the six month (17.2% vs. 13.3%) follow-up group. Conclusions: RA and S-IVL could be safe and effective therapeutic strategies for calcified LM disease. Further studies with a higher number of participants and longer follow-up times are warranted to establish the potential benefits of RA and S-IVL for the management of LM stenosis.

## 1. Introduction

Calcified Left Main (LM) stenosis represents a high-risk subset for percutaneous coronary intervention (PCI), and it has been recognized as a marker for increased rates of periprocedural complications and worse long-term outcomes [1,2]; therefore, according to the current revascularization guidelines [3], the coronary artery bypass grafting (CABG) remains as the preferred revascularization strategy. However, in clinical practice, the high surgical risk and coexisting comorbidities require the PCI to function as a reasonable alternative for selected patients. The severity of coronary calcification is a strong marker for unfavorable PCI outcomes [4]; indeed, increasing the probability of suboptimal stent expansion could lead to higher rates of in-stent restenosis and stent thrombosis. Adequate lesion preparation plays an indispensable role in the prevention of these adverse events, and it facilitates optimal stent delivery and expansion.

Recently, some advanced technologies and sophisticated techniques have been introduced to the PCI armamentarium in order to achieve adequate lesion preparation with a calcium crack [5,6,7]. Following its introduction to clinical practice, the rotational atherectomy (RA) has become the most commonly used tool to modify severely calcified plaques. The subpopulation of patients with unprotected heavily calcified LM disease is often underestimated in studies that have a high number of participants, as they are focused on the performance of debulking devices [8,9,10]. Along with the fact that most of those subjects are considered to be extremely high-risk in terms of the PCI results, there is a lack of strong evidence related to the efficacy of this kind of PCI intervention in the LM disease setting.

In contrast to the well-established RA technology [11,12], recently, a novel balloon-based coronary system—Shockwave C2 Intravascular Lithotripsy (S-IVL) (Shockwave Medical Inc., Santa Clara, CA, USA)—that relies on pulsatile mechanical energy, has been proposed to help manage the heavily calcified lesions. S-IVL leads to the profound defragmentation of calcium deposits without interacting with other vascular layers. The evidence for the efficacy and safety of S-IVL is mostly provided by the studies that were conducted on peripheral artery atherosclerosis [13], and real-life registries [14] which have evidence pertaining to its efficacy for LM disease are scarce [15].

Therefore, we designed this study to compare the efficacy and safety of intravascular lithotripsy and rotational atherectomy in the treatment of calcified unprotected Left Main stenosis.

## 2. Materials and Methods

The study presents a retrospective analysis of a dataset from a registry conducted in two cooperative, high-volume PCI cardiac centers (Department of Cardiology, The Copper Health Centre (MCZ), Lubin, Poland and Department of Cardiology, Provincial Specialized Hospital, Legnica, Poland). The subjects in this study were carefully selected from all consecutive patients with calcified lesions who had undergone PCI, and who required additional lesion preparation with either Rotational Atherectomy or Shock Wave Intravascular Lithotripsy for LM diseases (main inclusion criteria to study). All procedures were performed between January 2014 and June 2021. The decision to perform the PCI was either based on a judgment made by the Heart Team or on a particular clinical indication (ongoing ischemia, lack of will for the alternative treatment options). All patients were thoroughly informed about all therapeutic options and PCI-related risks before providing written informed consent for the procedure. There were no vessel-related exclusion criteria regarding lesion anatomy, length, tortuosity, severity, or prior stent placement. From the analysis, we excluded patients who had undergone simultaneous RA and S-IVL during one PCI procedure. Figure 1 presents a flow chart of the study protocol, along with the inclusion and exclusion criteria.

All clinical decisions regarding the PCI procedure, including the use of RA or S-IVL, access point, the use of left ventricular supporting devices, guiding catheter size, intravascular imaging guidance (OCT/IVUS), burr or shock-wave balloon size, rotablation speed, number of ultrasonic pulses applied, glycoprotein IIb/IIIa inhibitors or catecholamines administration, and bifurcation stenting technique, were determined at the operators’ discretion.

All the RA procedures were performed by experienced operators who had previously performed at least 50 RA-related PCI procedures. RA procedures were performed in accordance with contemporary clinical practice. In brief, burr size selection was left to the operators’ to decide; however, all of them were encouraged to reach a burr/vessel ratio of 0.5 [16]. Rotational velocity was set at 140,000–170,000 rpm, taking special care to avoid rotational speed decelerations of >5000 rpm. During the intervention, all patients received intravenous unfractionated heparin (UFH) (70–100 IU/kg), along with a continuous intracoronary infusion of verapamil, nitroglycerin, and UFH, which was used as a slow-flow prevention cocktail.

The LM Shock-Wave intravascular lithotripsy procedures were performed by high-volume PCI operators (at least 200 PCI procedures conducted per year) who completed full training with the Shock Wave device. Details concerning technique when using the S-IVL (previous predilatation, size of the catheter, number of pulses) were left to the discretion of each operator, but they were strongly encouraged to properly size the diameter of the S-IVL catheter (sized 1:1 to the reference vessel diameter). During all the interventions, intravenous UFH (70–100 IU/kg) was used as an antithrombotic drug.

The data was retrospectively collected, including the angiographic and procedural characteristics, initial clinical characteristics, as well as clinical follow-up characteristics (30 days and 6 months post-discharge). The clinical follow-up was performed by physicians, and it focused on the analysis of hospital records and out-hospital scheduled visits made by the specialist, along with telephone contact which was conducted after the aforementioned periods of time. There were no vessel-related exclusion criteria regarding lesion anatomy, length, tortuosity, severity, or prior stent placement. Of the consecutive 45 cases that were analyzed, we excluded only one subject, who required the use of both lesion preparation techniques (RA and S-IVL) during the same procedure.

The primary endpoint of the study was the occurrence of in-hospital major adverse cardiac events (MACE). The MACE was composed of incidents such as death, myocardial infarction, an urgent need for target vessel revascularization, and probable or definite stent thrombosis. The secondary endpoints included the occurrence of MACE at 6 months, cerebrovascular episodes, all kinds of revascularization procedures, and scaffold restenosis. Myocardial infarction was defined according to the Fourth Universal Definition of Myocardial Infarction [17]. Target vessel revascularization was defined as any repeated percutaneous intervention or surgical bypass of any segment of the target vessel, including the target lesion [18]. Stent thrombosis was defined as the presence of a thrombus that originated in the stent or in the segment, and was 5 mm proximal or distal to the stent; alternatively, the thrombus could emerge from a side branch originating from the stented segment, coupled with the presence of at least one of the criteria for ongoing ischemia (acute onset of ischemic symptoms at rest or/and new electrocardiographic changes that are suggestive of acute ischemia or/and a typical rise and fall in cardiac biomarkers) [18].

All the statistical analyses were performed using the R language version 4.0.4 (R Foundation for Statistical Computing, Vienna, Austria) [19]. Continuous variables were characterized using their mean and standard deviation if the variables had a normal distribution; otherwise, median and interquartile ranges were used and presented in square brackets. Frequencies were used for categorical variables. The differences between the means were assessed using the student’s *t*-test or the Mann–Whitney U test, depending on the distribution of the variables and the variety of variances, as previously assessed using the Shapiro–Wilk test and Levene’s test. The Fisher’s exact test was used for categorical variables. *p*-values < 0.05 were accepted as a threshold for statistical significance.

## 3. Results

During the study period, 45 consecutive patients who had undergone LM-PCI, and were supported by RA or S-IVL, were enrolled. Exemplar PCI procedures are provided in Figure 2. One subject required the use of both techniques, and therefore, they were excluded from the analysis. The baseline clinical characteristic of both study groups is detailed in Table 1; additionally, the Appendix A contain the study-related STROBE (Strengthening The Reporting of OBservational Studies in Epidemiology) checklist.

The RA group consisted of 29 patients who were predominantly male (72.4%) with ACS (55.2%). The mean age was 70.3 ± 9.1. Similarly, the majority of the subjects in the S-IVL group were male (80.0%), with a mean age of 72.1 ± 6.1 years, and with a relatively high prevalence of ACS cases (60.0%). We noticed a high prevalence of cardiovascular risk factors among both groups. Despite this, no significant differences in clinical features between the study arms were noted, though a trend indicating a higher prevalence of previous coronary artery disease, including for those who had previously undergone MI, CABG, and PCI procedures, was observed in the Rota group.

All the data regarding procedural features were pooled in Table 2. When analyzing the initial advancement of coronary artery disease, a significantly higher syntax score value was observed in the RA arm (28.0 ± 7.5 vs. 23.3 ± 13.0; *p* = 0.038). Interestingly, Syntax II score values showed an opposite trend; however, they do not have any statistical significance. Patients in the RA group had a lower prevalence of radial access, and they also used the 7F and 8 F guiding catheters during the PCI procedures more frequently. The subjects assigned to the RA group were characterized by a less aggressive initial lesion predilatation (size of balloon catheter 3.24 [3–3.5] mm vs. 2.78 [2.5–3.0] mm *p* = 0.002; predilatation pressure 19.2 ± 1.4 atm vs. 21.2 ± 1.3 atm *p* = 0.031) compared with the S-IVL group.

The mean S-IVL balloon size was 3.39 ± 0.40 mm, with an average pulse number of 44.28 ± 27.9. In the RA group, the mean final burr size was 1.51 ± 0.18 mm, and only two subjects required burr size enhancement. The average rotational speed was set at 166.875 ± 4787 rpm, with a mean rotablation duration time of 241 ± 129.1 s.

There were no significant differences between the data collected in-hospital and during the six month follow-up. All the outcomes from the data are presented in Table 3.

In the RA group, we recorded three in-hospital deaths. One death was periprocedural—patient with prehospital cardiac arrest—and the procedure was carried out with Lucas (Stryker Medical, Portage, MI, USA) CPR assist device support. Two deaths occurred post-procedure (patients were transferred after the PCI to a local ICU due to multiorgan dysfunction; the deaths were reported 10 and 17 days after the index procedure); we noticed one death occurred for an unknown reason, 17 days after discharge, in a patient with multiple comorbidities and who suffered from alcohol abuse. Moreover, one MI involving a previously untreated vessel occurred in the RA group 120 days after the index procedure. Furthermore, we observed three periprocedural vessel perforations. One LAD rupture, with acute cardiac tamponade, was managed with urgent pericardiocentesis, and two others were caused by distal wire vessel damage without hemodynamic consequences.

In the S-IVL subpopulation, one fatal in-stent thrombosis occurred during the in-hospital period (five days after a PCI). Additionally, post-discharge (14 days after the procedure), we observed a second death in the S-IVL group. A subject with numerous comorbidities and low LVEF (25%) died for unknown reasons, and the patient was under clinical observation before the scheduled ICD implantation.

## 4. Discussion

To the best of our knowledge, this is the first pilot study to be conducted in humans that evaluates the safety and efficiency of two calcium modification devices—Rotational Atherectomy and Shock Wave Intravascular Lithotripsy in subjects with LM diseases.

The main finding of our observational study is that both the Rotational Atherectomy and Shock Wave Intravascular Lithotripsy devices seem to be practical, safe, and effective in the management of calcified lesions in the Left Main, as demonstrated through both short- and mid-term follow-ups.

Patients with LM disease, that are undergoing PCI, are considered to be a high-risk population. Large myocardial ischemic areas, along with potential technical difficulties during a PCI procedure, are reflected in the current guidelines which support CABG as the preferred revascularization method [20]. In particular, significant calcifications are associated with a higher incidence of periprocedural complications, impeded stent delivery, and issues with appropriate expansion which affects both the short- and long-term outcomes [21]. Despite that fact, PCI often remains the last therapeutic option for patients who are not suitable for surgery.

Interestingly, a higher Syntax score was observed in the Rotational Atherectomy cohort when compared with the S-IVL cohort. Surprisingly, the opposite tendency was observed in terms of the Syntax II score; however, it was not of statistical significance. This might be due to the non-randomized, retrospective character of the study. It might also partially be the result of an unintended selection of patients. Due to the advantages and limitations of both methods, patients with longer, more diffused lesions were more prone to be assigned to the RA arm, whereas patients with focal, eccentric lesions might have been more likely to be assigned to the S-IVL cohort. Nevertheless, the preliminary character of this study, and the relatively low sample size, indicate the need for subsequent, larger, prospective randomized trials which precisely evaluate the safety and efficiency of both methods in terms of the management of LM disease.

What needs to be emphasized, is that both analyzed groups represent a unique, ultra-high-risk subset in current interventional cardiology. The relatively high Syntax Score I and Syntax Score II in both arms of the study support this thesis. The use of PCI for these challenging cases is constantly growing, which is largely due to novel devices that facilitate adequate lesion preparation and improvements to stent technologies.

Nevertheless, PCI continues to be technically challenging, and the outcomes are unsatisfactory; however, despite this, in our cohort, patients after the RA procedure had a reasonable six month all-cause mortality level of 13.7%, and a cumulative MACE rate of 17.2%. These data appear to be consistent with the data previously reported by Yabushita et al. [22] (one year all-cause mortality level of 9.3%, with a subsequent MACE ratio of 23.4%), Garcia et al. [23] (in-hospital mortality level of 7.5%, compared with the 10.3% that was observed in the present study), and Ileasi et al. [24] (one year mortality rate of 12.6%, and a MACE rate of 26.4%). Nevertheless, unlike the aforementioned registries, we performed procedures on the majority of cases in the ACS subset. Due to thrombotic issues and potential hemodynamic or electrical instability, the risk of adverse clinical events is increased [25,26].

Interestingly, when comparing the RA-LM- PCI and S-IVL-LM-PCI results, we noticed an insignificant trend wherein the MACE rate increased (17.2% vs. 13.3%), with a similar all-cause mortality level. There is a lack of strong evidence supporting the efficiency of S-IVL-LM-PCI; however, our results partially confirm some other trials [27,28,29,30]. These findings might be linked to the decreased rate of periprocedural complications after S-IVL. Relatively lower rates of perforation were observed, and the slow/no-flow phenomena have been confirmed in the non-LM studies [31,32,33], which are also associated with the mechanism of action in S-IVL. Since lithotripsy leads to defragmentation and fractures of the calcific plaque without affecting a vessel wall, it does not generate the plaque ablation, meaning that the occurrence of the no-flow phenomenon or perforation is very unlikely.

Additionally, a strong relationship between the method applied for lesion preparation and the vascular access point was observed. Femoral access with a large size for guiding catheters was the preferred approach for the LM-RA, whereas the majority of S-IVL cases (80%) were performed via radial access. Prior studies [34,35,36] have shown that radial access, compared with femoral access, provides a similar efficacy profile while increasing procedural safety features; this occurs largely as a result of the decreased rate of major bleeding and access point complications. Notably, S-IVL, compared with RA, is a relatively simple therapy with a shorter learning curve, and importantly, due to safety concerns, throughout the lithotripsy procedure, the LM-PCI allows the maintenance of a buddy wire in the side branch. Interestingly, the use of Rotational Atherectomy does not exclude the use of Shockwave Intravascular Lithotripsy (one case of RA combined with S-IVL was excluded from the study due to methodological issues), which could constitute a novel therapeutic option, and it has been recently proposed as a novel bail-out strategy for extremely resistant calcified coronary lesions [7,37].

Although the prevalence of intravascular imaging support use in our study cohort is above average in general practice [38,39], it is still lower than average in the Left Main Disease PCI subset [40]. This is probably related to several factors. Firstly, the vast majority of cases in both study cohorts were performed in ASC conditions, often as a bailout treatment due to the failure of classic balloon-dependent methods. Such an urgent subset is often related to thrombotic issues [41,42]; therefore, operators tend to keep the procedure as simple as possible. Many of the procedures were performed in critical Left Main lesions that were connected with the large myocardial ischemic territory; this additionally hindered the preliminary assessment of the lesion with intravascular imagining devices and could thus potentially affect the outcomes of our study.

It has been well documented that an initial assessment of culprit lesions in intravascular imaging [43,44], with subsequent PCI guidance [45,46], can reduce the rate of unfavorable clinical outcomes. Presumably, the data obtained from intravascular imaging, including the IVUS and/or OCT analyses, would provide valuable and precise information regarding the nature of the calcium burden and evidence of calcium modification, along with a comparison of the acute luminal gain achieved by RA compared with S-IVL. The relationship between this data and different mechanisms of plaque modification using both devices [47,48,49] could result in the evaluation of new criteria for optimal subjects who qualify for the primary Left Main Rotablation or/and S-IVL; therefore, we strongly believe that future studies which foreground this issue are of urgent clinical need.

### Limitations

This was a non-randomized retrospective observational pilot study with a relatively short observation period (six month follow-up). The study population was not large, and it was underpowered in terms of its ability to conduct a reliable assessment of events. Moreover, the rate of intravascular guidance for PCI procedures was comparatively low.

## 5. Conclusions

Rotational Atherectomy and Shock Wave Intravascular Lithotripsy appear to be safe and effective therapeutic options for the management of calcified LM lesions. Further studies with a higher number of participants, a longer observation period, and broader support for intravascular imaging are required to establish the potential benefits of RA or S-IVL in the treatment of the LM stenosis.

## Figures and Tables

**Figure 1 ijerph-19-09011-f001:**
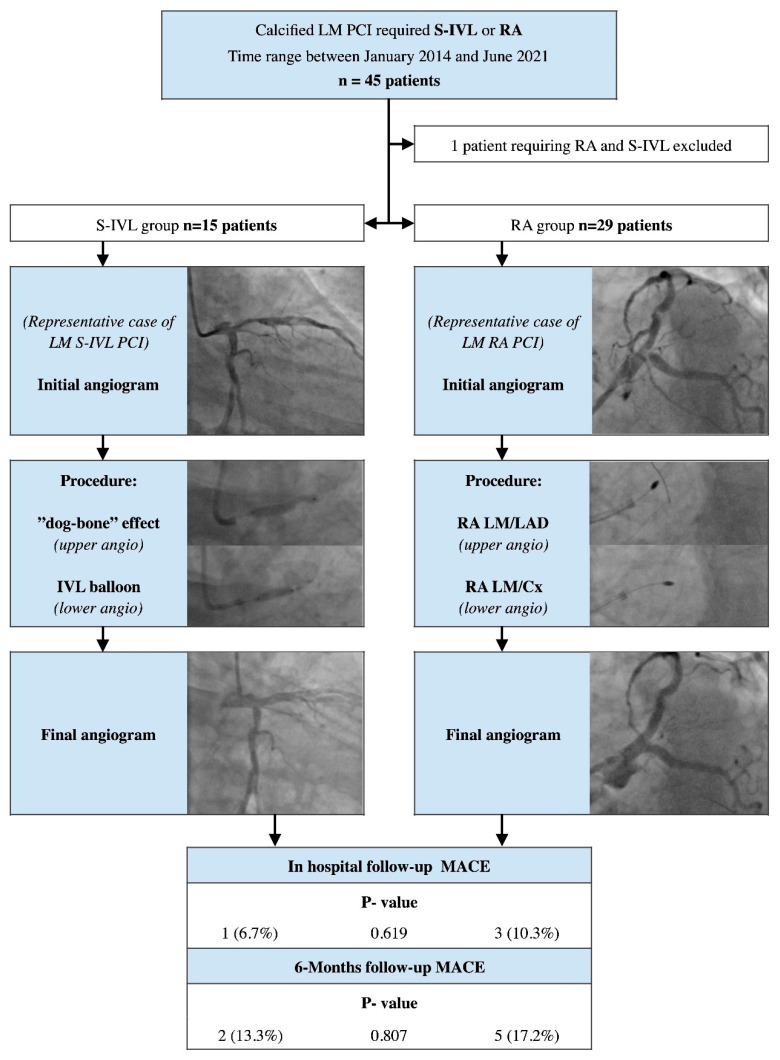
A flow chart of the study protocol with inclusion and exclusion criteria. Abbreviations: LM—Left Main; PCI—percutaneous coronary intervention; S-IV—Shock Wave Intravascular Lithotripsy; RA—Rotational Atherectomy; LAD—Left Anterior Descending artery; Cx—Circumflex artery; MACE—major adverse cardiac events.

**Figure 2 ijerph-19-09011-f002:**
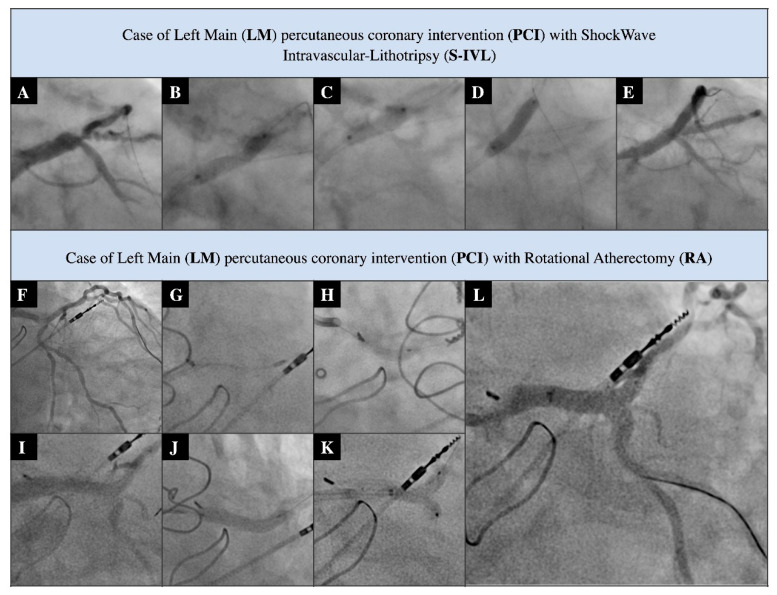
Exemplary PCI procedures. S-IVL-related procedure (**A**–**E**). (**A**) Distal LM significant stenosis; (**B**) non-compliant 3.5 mm balloon under-expansion; (**C**) S-IVL 3.5 mm balloon inflation; (**D**) stenting with DES 4. 0 mm; and (**E**) final post PCI angiogram. RA-related procedure (**F**–**L**). (**F**) Distal LM critical stenosis with severe calcifications; (**G**) RA with burr 1.75 mm; (**H**) predilatation of LM with 3.5 mm non-compliant balloon; (**I**) pre-stenting angiogram; (**J**) LM/LAD stenting with DES 4.0 × 20 mm; (**K**) LM/LAD/Cx kissing balloons with non-compliant 3.5 mm/3.0 mm balloons; and (**L**) final post-PCI angiogram.

**Table 1 ijerph-19-09011-t001:** The baseline clinical characteristics of both study groups.

	Rotational Atherectomy (RA) N-29	Shockwave Intravascular (S-IVL) N-15	*p*-Value
Age	70.3 ± 9.1	72.1 ± 6.1	0.478
Gender male (ratio)	21 (72.4%)	12 (80.0%)	0.532
Stable angina	13 (44.8%)	6 (40.0%)	0.591
Unstable angina	6 (20.7%)	2 (13.3%)	0.511
NSTEMI	9 (31.0%)	6 (40.0%)	0.535
STEMI	1 (3.4%)	1 (6.7%)	0.561
Diabetes mellitus	16 (55.2%)	10 (66.7%)	0.487
Chronic heart failure	9 (31.0%)	6 (40.0%)	0.412
Hypertension	25 (86.2%)	13 (86.7%)	0.591
Hyperlipidemia	20 (68.9%)	15 (100%)	0.293
Atrial Fibrillation	5 (17.2%)	5 (33.3%)	0.248
History of PCI	16 (55.2%)	7 (46.7%)	0.544
History of MI	13 (44.8%)	5 (33.3%)	0.360
History of CABG	6 (20.7%)	1 (6.7%)	0.311
COPD	6 (20.7%)	2 (13.3%)	0.511
Chronic kidney diseases	13 (44.8%)	6 (40.0%)	0.519

Abbreviations: NSTEMI—no ST-Elevation Myocardial Infraction; STEMI—ST-Elevation Myocardial Infraction; PCI—percutaneous coronary intervention; MI—Myocardial Infraction; CABG—coronary artery bypass grafting; COPD—Chronic Obstructive Pulmonary Diseases.

**Table 2 ijerph-19-09011-t002:** The baseline procedural features of both study groups.

	Rotational Atherectomy N-29	Shockwave Intravascular N-15	*p*-Value
Syntax I score	28.0 ± 7.5;	23.3 ± 13.0	**0.038**
Syntax II—PCI score	35.8 ± 8.4	38.7 ± 14.8	0.489
Syntax II PCI four year mortality	9.6 [7.7–15.2]	10.1 [5–34]	0.876
Syntax II—CABG score	34.5 ± 9.0	38.3 ± 10.5	0.175
Syntax II CABG year mortality	11.5 [6.8–28.9]	10.3 [6–20.1]	0.414
Radial Access	15 (51.7%)	12 (80.0%)	0.287
6F Guide Catheter	3 (10.3%)	4 (26.7%)	0.115
7F or larger Guide Catheter	26 (89.6%)	11 (73.3%)	0.263
Predilatation	26 (89.6%)	15 (100%)	0.498
Predilatation balloon diameter (mm)	2.78 [2.5–3.0]	3.24 [3–3.5]	**0.002**
Predilatation pressure (atm)	21.2 ± 1.4	19.2 ± 1.4	**0.031**
Single stent technique	21 (72.4%)	11 (73.3%)	0.458
Two stent bifurcation technique	8 (27.6%)	3 (20.0%)	0.357
Postdilatation—POT	26 (89.6%)	14 (93.3%)	0.558
Intravascular Guidance	3 (10.3%)	3 (20.0%)	0.420
Perforation	3 (10.3%)	0 (0%)	0.327
No-flow phenomenon	1 (3.4%)	0 (0%)	0.561
Administration of catecholamines	2 (6.9%)	2 (13.3%)	0.420
Acetylsalicylic Acid	29 (100%)	15 (100%)	1
Clopidogrel	19 (65.6%)	9 (60%)	0.552
Ticagrelol	10 (34.4%)	6 (40%)	0.552

Abbreviations: PCI—percutaneous coronary intervention; CABG—coronary artery bypass grafting; POT—proximal optimization technique; bold text—statistically significant value.

**Table 3 ijerph-19-09011-t003:** Clinical follow-up data of both groups.

	Rotational Atherectomy N-29	Shockwave Intravascular N-15	*p*-Value
In hospital follow-up			
MACE	3 (10.3%)	1 (6.7%)	0.619
Death	3 (10.3%)	1 (6.7%)	0.619
Myocardial infarction	0 (0%)	1 (6.7%)	0.341
Target vessel revascularization	0 (0%)	1 (6.7%)	0.341
Stent thrombosis	0 (0%)	1 (6.7%)	0.341
Cerebrovascular episodes	0 (0%)	0 (0%)	–
Stent restenosis	0 (0%)	0 (0%)	–
Any revascularization	0 (0%)	0 (0%)	–
Six Month follow-up			
MACE	5 (17.2%)	2 (13.3%)	0.807
Death	4 (13.7%)	2 (13.3%)	0.965
Myocardial infarction	1 (3.4%)	1 (6.7%)	0.619
Target vessel revascularization	0 (0%)	1 (6.7%)	0.151
Stent thrombosis	0 (0%)	1 (6.7%)	0.151
Cerebrovascular episodes	0 (0%)	0 (0%)	–
Stent restenosis	0 (0%)	0 (0%)	–
Any revascularization	2 (6.9%)	0 (0%)	0.413

Abbreviations: MACE—Major adverse cardiac events.

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
