# Peer review of "Intravascular Lithotripsy as a Novel Treatment Method for Calcified Unprotected Left Main Diseases—Comparison to Rotational Atherectomy—Short-Term Outcomes"

_ijerph, 2022, doi:10.3390/ijerph19159011_

Round 1

Reviewer 1 Report

Nice retrospective analysis of clinical and procedural outcomes using rota vs IVL to treat calcific LMCA disease.

The main concern I have regarding this study is the relatively low use of intravascular imaging which is recommended by current guidelines.  Intravascular imaging data for the small subset of patients in which it was performed should have been included.  The readers would be interested to understand the nature of calcium burden as defined by arc of calcium, length and thickness of calcium, as well as evidence of calcium modification and/or fracture but rota vs IVL.  Finally, a comparison of acute luminal gain achieved by rota vs IVL would be very helpful given the long term prognostic implications.

Author Response

We would like to thank the Reviewer for an in-depth analysis of the manuscript and for the pivotal comments provided, which have resulted in a significant improvement of this manuscript.

Comment 1 

We would like to thank the Reviewer for the remark regarding the relatively low rate of intravascular imaging in our study. We agree with the opinion that it is an important issue (we mentioned it in the limitation section) therefore we decided to make a comment focused on this topic in discussion section.

Although the prevalence of intravascular imaging support in our study cohort is above average use in general practice [39,40], still it is lower than average in the Left Main Diseases PCI subset [41]. This fact is probably related to several factors. Firstly, the vast majority of cases in both study cohorts were performed in ASC – conditions, often as a bailout treatment due to the failure of classic balloon-depend methods. Such an urgent subset is often related to thrombotic issues [42,43] therefore operators tend to keep the procedure as simple as possible. Many of the procedures were performed in critical Left Main lesions connected with the large myocardial ischemic territory which additionally hindered the preliminary assessment of the lesion with intravascular imagining devices. This fact could potentially affect the outcomes of our study.

It has been well documented that initial assessment of culprit lesion in intravascular imaging [44,45] with subsequent PCI guidance [46,47] can reduce the rate of unfavorable clinical outcomes. Presumably, the data obtained from intravascular imaging, including the IVUS and/or OCT analyses, would provide valuable and precise information regarding the nature of calcium burden, evidence of calcium modification, along comparison of acute luminal gain achieved by RA compare to S-IVL. The relationship of this data with different mechanisms of plaque modification by both devices [48,49,50] could result in the evaluation of new criteria for optimal subjects qualifying for the primary Left Main Rotablation or/and S-IVL. Therefore we strongly believe that future studies key to this issue is an urgent clinical need. “

Furthermore, we added also a short comment regarding this topic in the conclusion section

Reviewer 2 Report

Calcified left main diseases still represents a challenge for clinical physicians, considering the increased rate of periprocedural complications and worse long-term outcomes. This is generally a well-written and comprehensive article that aimed to compare the efficacy and safety of intravascular lithotripsy and rotational atherectomy in the treatment of calcified unprotected left main stenosis. I consider that the study is valuable and has interesting findings that can have important implications for clinical practice. In order to improve the quality of the study, I have some suggestions:

1. Please insert some coronarographic images of the patients included in the study.

2. I consider that it is important to address the future scope and topics that are important and that could not be covered in the manuscript.

Author Response

We would like to thank the Reviewer for an in-depth analysis of the manuscript and for the pivotal comments provided, which have resulted in a significant improvement of this manuscript.

Comment 1 We would like to thank the Reviewer for pointing out fact that the manuscript contains a relatively low number of pictures of LM-PCI (only a few in figure 1) therefore we decided to add an additional Figure 2.

Comment 2  We believe that the Reviewer made a valid point asking about the future scope and topics that are important and that could not be covered in the manuscript. We added in the discussion section. “ It has been well documented that initial assessment of culprit lesion in intravascular imaging [44,45] with subsequent PCI guidance [46,47] can reduce the rate of unfavorable clinical outcomes. Presumably, the data obtained from intravascular imaging, including the IVUS and/or OCT analyses, would provide valuable and precise information regarding the nature of calcium burden, evidence of calcium modification, along comparison of acute luminal gain achieved by RA compare to S-IVL. The relationship of this data with different mechanisms of plaque modification by both devices [48,49,50] could result in the evaluation of new criteria for optimal subjects qualifying for the primary Left Main Rotablation or/and S-IVL. Therefore we strongly believe that future studies key to this issue is an urgent clinical need. “

Furthermore, we added also a short comment regarding this topic in the conclusion section

Reviewer 3 Report

I read with interest the manuscript entitled "Intravascular Lithotripsy as a novel treatment method for calcified unprotected left main diseases – comparison to Rotational 3 Atherectomy - short-term outcomes." by Riola et al. It deals with an interesting and contemporary topic; rotational atherectomy and intravascular shockwave lithotripsy in patients with left main disease. Importantly, a significant proportion of their patient population presented with an ACS. Below are my comments:

1) While the involved scientific field is of great importance, it is unclear what this study can add due to the extremely small sample size. Even though the authors have acknowledged this limitation, it downgrades significantly the importance of the study's findings.

2) Keywords: Please remove the novel treatment method keyword. Also, the CHIP abbreviation does not seem to match the high-risk percutaneous coronary intervention, please explain.

3) Methods: Please mention the name of the centers and laboratories.

4) Methods: Please include the inclusion and exclusion criteria in the text.

5) Line 90: The authors mentioned that experienced operators performed the RA procedures. Those 50 RA-related PCIs were performed prior to the study period?

6) Line 102: The abbreviation of UFH is being defined for a second time there. 

7) Line 125: What version of R language/studio was used?

8) Lines 129-130: Did you analyze both normally and non-normally distributed variables with Mann-Whitney? If so, I think that normally distributed continuous variables should be analyzed with t-test.

9) I think that the strobe checklist should be followed, please include it as a supplementary material and also mention it in the text. 

10) Table 2: p-values should be more adequately aligned, please adjust.

11) Please remove the unit of measurement after the mean, as it is present after the standard deviation. Correct throughout the text.

Author Response

We would like to thank the Reviewer for an in-depth analysis of the manuscript and for the pivotal comments provided, which have resulted in a significant improvement of this manuscript.

Comment 1  We believe that the Reviewer made a valid pointing out a relatively small size sample in our study. However, what needs to be emphasized subpopulation of patients with heavily calcified Left Main disease requiring the use of RA or other debulking devices is a rare clinical subset. All of these subjects are at extremely high risk of PCI. The high-numbers Rotational Atherectomy registries support these findings [1,2]. Therefore the data regarding this clinical issue are mainly provided by small non-randomized studies similar to our study either multicenter observational registries (with all the methodological imperfections of this type of research). Due to the novelty of the S-IVL device clinical data regarding the LM interventions are scar. Even more, studies comparing LM intentions with the use of RA and S-IVL are missing. Therefore, we believe that despite the methodological shortcomings of our study it is valuable from a clinical point of view, particularly for readers who are the first-line practitioners in Cardiovascular Interventions.

  1. Iannaccone M, Barbero U, D'ascenzo F, Latib A, Pennacchi M, Rossi ML, Ugo F, Meliga E, Kawamoto H, Moretti C, Ielasi A, Garbo R, Colombo A, Sardella G, Boccuzzi GG. Rotational atherectomy in very long lesions: Results for the ROTATE registry. Catheter Cardiovasc Interv. 2016 Nov 15;88(6):E164-E172. doi: 10.1002/ccd.26548. 
  2. Januszek R, Siudak Z, Dziewierz A, Rakowski T, Legutko J, Dudek D, Bartuś S. Bailout rotational atherectomy in patients with myocardial infarction is not associated with an increased periprocedural complication rate or poorer angiographic outcomes in comparison to elective procedures (from the ORPKI Polish National Registry 2015-2016). Postepy Kardiol Interwencyjnej. 2018;14(2):135-143. doi: 10.5114/aic.2018.76404. 

Comment 2 We believe that the Reviewer made a valid point asking to change the keywords of the manuscript – we performed it due to mentioned recommendations.

Comment 3. Followed by the Reviewer suggestion we added data of centers where PCI procedures were carried out in methodology section.

Comment 4 –We would like to thank the Reviewer for a suggestion regarding adding inclusion and exclusion criteria into the body of manuscript. We added the necessary data in the Material and Methods section.

Comment 5 We are pleased to inform you that all operators performed at least 50 RA prior to study.

Comment 6 – We would like to thank the Reviewer for finding an editorial mistake in the body of the manuscript we corrected it.

Comment 7 We would like to thank the reviewer for pointing out the missing version of the R language. For the purpose of the analyses presented in this manuscript the R version 4.0.4 was used, which we have now specified in the appropriate part of the Methods section.

Comment 8 We agree with the Reviewer, that the continuous variables with normal distribution and homogenous variation should be compared using the T-test. Indeed, in our study, the differences between the means were assessed using the student's T-test or the Mann-Whitney U test, depending on the distribution of the variables and the variety of variances, assessed using the  Shapiro-Wilk test and Levene’s test as appropriate. In order to be more precise, we have commented on it the Methods section as following:

All the statistical analyses were performed using the R language version 4.0.4  [16]. Continuous variables were characterized with their mean and standard deviation if variable has a normal distribution, otherwise median and interquartile range were used and presented in square brackets. Frequencies were used for categorical variables. The differences between the means were assessed using the student's T-test or the Mann-Whitney U test, depending on the distribution of the variables and the variety of variances, as assessed before using the Shapiro-Wilk test and Levene’s test. The Fisher's exact test was used for categorical variables. P-values < 0,05 were accepted as a threshold for statistical significance.

Comment 9. As the Reviewer suggested, we followed the STROBE checklist and added the necessary file to supplementary files we also provide a necessary statement in the material and methods section.

Comment 10 We believe that the Reviewer made a valid point while stating the p values in the Table 2 should be aligned more precisely. As result, we have corrected the data presented in the Table, according to this suggestion.

Comment 11  We believe that the Reviewer made a valid point asking to unify data presentation in the manuscript we performed suggested changes.

Round 2

Reviewer 1 Report

Revisions adequately address original critique. 

Reviewer 3 Report

No further comments.